# ERα activity depends on interaction and target site corecruitment with phosphorylated CREB1

Melissa Berto[1], Valerie Jean[1], Wilbert Zwart[2], Didier Picard[1]

The two transcription factors estrogen receptor α (ERα) and cyclic adenosine monophosphate (cAMP)–responsive element binding protein 1 (CREB1) mediate different signals, bind different response elements, and control different transcriptional programs. And yet, results obtained with transfected reporter genes suggested that their activities may intersect. We demonstrate here that CREB1 stimulates and is necessary for ERα activity on a transfected reporter gene and several endogenous targets both in response to its cognate ligand estrogen and to ligand-independent activation by cAMP. The stimulatory activity of CREB1 requires its DNA binding and activation by phosphorylation, and affects the chromatin recruitment of ERα. CREB1 and ERα are biochemically associated and share hundreds to thousands of chromatin binding sites upon stimulation by estrogen and cAMP, respectively. These shared regulatory activities may underlie the anti-apoptotic effects of estrogen and cAMP signaling in ERα-positive breast cancer cells. Moreover, high levels of CREB1 are associated with good prognosis in ERα-positive breast cancer patients, which may be because of its ability to promote ERα functions, thereby maintaining it as a successful therapeutic target.

## Introduction

Estrogen receptor α (ERα) and cAMP-responsive element binding protein 1 (CREB1) are two unrelated transcription factors that, at first sight, have nothing to do with each other. ERα is a member of the nuclear receptor superfamily; in response to binding its cognate ligand estrogen, it is activated as a transcription factor and binds as a homodimer to specific DNA sequences across the genome to regulate target genes (Heldring et al, 2007; Flach & Zwart, 2016). CREB1 and other family members such as ATF1 contain a basic region for binding DNA followed by a leucine zipper for homo- or heterodimerization; as its name indicates, it is a paradigmatic target of the cAMP-activated PKA. Upon phosphorylation of serine 133 by PKA, pCREB1 can specifically recruit the coactivator CREB binding protein (CBP) and its paralog p300 (Mayr & Montminy, 2001). Thus, whereas ERα is typically a ligand-activated transcription factor, CREB1 is signal responsive through phosphorylation. Intriguingly, unliganded ERα can also be activated by cAMP signaling in the absence of estrogens (Power et al, 1991; Aronica & Katzenellenbogen, 1993; Smith et al, 1993; Lazennec et al, 2001; Dudek & Picard, 2008; Carascossa et al, 2010; de Leeuw et al, 2013). We had found this extreme form of signaling crosstalk to be dependent on the PKA-mediated phosphorylation of the coregulators CARM1 (Carascossa et al, 2010) and LSD1 (Bennesch et al, 2016), but could not exclude that yet other factors might be involved (Bennesch & Picard, 2015).

CREB1 not only controls the expression of its own direct target genes (Mayr & Montminy, 2001; Zhang et al, 2005), but is also involved in signaling crosstalk with nuclear receptors such as the glucocorticoid receptor (GR) (Akerblom et al, 1988) and ERα (Lazennec et al, 2001). Whether CREB1 stimulates or represses nuclear receptor activity seems to be cell-context dependent (Lazennec et al, 2001; Diaz-Gallardo et al, 2010; Ratman et al, 2013). Similarly, ERα interacts with a variety of transcription factors (Heldring et al, 2007, 2011), be it by tethering to them on their target sites, such as in the case of the Jun/Fos heterodimer AP-1 (Kushner et al, 2000) and SP1 (Saville et al, 2000), or through a variety of other, not always well-characterized mechanisms. ERα and the NF-κB display both positive and negative interactions (Kalaitzidis & Gilmore, 2005; Franco et al, 2015), as do ERα and the retinoic acid receptors at the level of chromatin binding (Hua et al, 2009; Ross-Innes et al, 2010). The crosstalk between GR and ERα may be highly context dependent because GR has been demonstrated to repress some of the ERα program by disrupting its transactivation complexes (Yang et al, 2017) and to stimulate some ERα responses by promoting chromatin remodeling such that ERα loading is facilitated (Voss et al, 2011; Swinstead et al, 2016a).

In view of the fact that both ERα and CREB1 mediate PKA-mediated cAMP signaling and considering previous reports on their crosstalk (Lazennec et al, 2001; Heldring et al, 2011), we decided to explore the mechanism and the physiological or pathological significance of signaling crosstalk in more detail. Because ERα is a key proliferative factor in breast cancer, we chose ERα-positive breast cancer cells as cellular model system.

[1]Département de Biologie Cellulaire and Institute of Genetics and Genomics of Geneva, Université de Genève, Genève, Switzerland   [2]Division of Oncogenomics, Oncode Institute, Netherlands Cancer Institute, Amsterdam, The Netherlands

Correspondence: didier.picard@unige.ch

# Results

## CREB1 stimulates the transcriptional activity of the liganded and unliganded ERα

To confirm and characterize further the positive contribution of CREB1 to ERα activity (Lazennec et al, 2001), we performed luciferase reporter assays with the human ERα-positive breast cancer cell line MDA-MB-134 (Reiner & Katzenellenbogen, 1986). Cells were transfected with luciferase reporters without an estrogen responsive element (Luc), with an estrogen responsive element (ERE-Luc), or with a CREB responsive element (CRE-Luc) (Fig 1). Cells were treated either with the physiological estrogen 17β-estradiol (E2) or with a cocktail of the adenylate cyclase activator forskolin and the phosphodiesterase inhibitor 3-isobutyl-1-methylxanthine (FI) to boost intracellular levels of cAMP. As expected (Dudek & Picard, 2008; Carascossa et al, 2010; Bennesch et al, 2016), ERα is transcriptionally activated by both treatments (Fig 1A); CREB1 is only activated by cAMP and E2-activated ERα cannot activate transcription through a CRE (Fig 1B). Without a response element, only basal activity could be observed (Fig 1C). Overexpression of wild-type CREB1 increases the activity of both ERα and CREB1 luciferase reporters, whereas overexpression of a dominant-negative form of CREB1 (A-CREB1), which is deficient in DNA binding because it lacks the basic region N-terminal to the leucine zipper dimerization domain (Ahn et al, 1998), reduces reporter activities (Fig 1A–C). Note that ERα levels are not affected by the overexpression of CREB1 (see below). Because CREB1 transcriptional activity is stimulated by phosphorylation of a conserved serine (Ser133) by cAMP-activated PKA, we tested whether this residue is important for the contribution of CREB1 to ERα activity. Therefore, we overexpressed either a phosphoserine-deficient mutant CREB1 (CREB1-S133A) or a phosphoserine-mimetic mutant

CREB1 (CREB1-S133D) (Fig S1A). The luciferase reporter assays of Fig 1 show that the non-phosphorylatable CREB1 mutant, acting as a dominant-negative mutant, reduces both ERα and CREB1 activities and that the phosphoserine-mimetic CREB1 mutant stimulates ERα similarly to wild-type CREB1. Conversely, when CREB1 was knocked down by RNA interference (Fig S1B–C), the luciferase activities of both reporters were decreased (Fig 1D). E2 does not seem to be able to promote the association of ERα with the DNA-bound CREB1 or to activate the CRE-Luc reporter; unlike what has been reported by others (Heldring et al, 2011), ERα is apparently unable to work by tethering under our experimental conditions. Note that based on experiments with two of the CREB1 mutants and exogenous expression of ERα, we conclude that CREB1 is also required for ERα function in ERα-negative HEK293T cells (Fig S2). Overall, these results indicate that CREB1 is both sufficient to boost ERα activity and necessary for its activity in response to either E2 or cAMP and that it is CREB1 itself, rather than other members of its family, that is involved.

Having demonstrated the importance of CREB1 for a transfected ERα reporter gene, we next evaluated this for endogenous ERα target genes (Fig 2). Overall, the overexpression of wild-type CREB1 or the CREB1 mutants A-CREB1 and S133A had the same effects on three representative ERα target genes (Fig 2A–C). The impact of the CREB1 mutant S133D on endogenous target genes induced by cAMP again recapitulated its effects on the transfected reporter gene, but S133D reduced the response to E2 for the three genes examined here. The latter finding may be a result of substantial differences in experimental conditions such as timing and context between the two types of experiments. Again, as for the transfected reporter, the knockdown of CREB1 dramatically reduced the activation of the ERα target genes NFKB2, TFF1, and GREB1 (Fig 2D). Remarkably, the expression of the two ERα target genes ABCA3 and NRIP1, which are only activated in response to E2, are only minimally affected by the CREB1 knockdown.

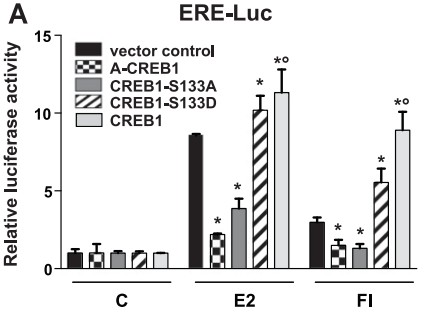

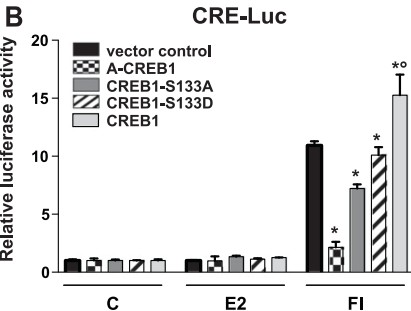

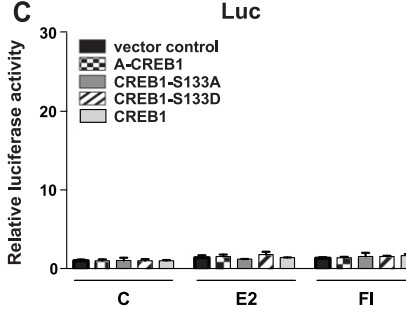

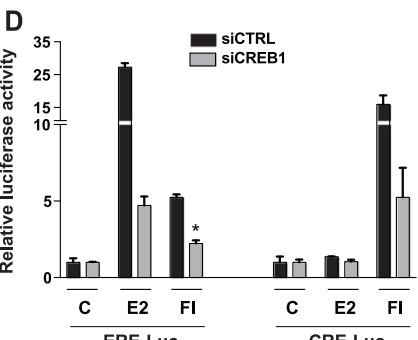

**Figure 1. CREB1 promotes ERα transcriptional activity.**
**(A–C)** The overexpression of exogenous CREB1 increases both ERα and CREB1 activities. Luciferase reporter assays in MDA-MB-134 breast cancer cells with endogenous ERα. Cells were cotransfected with a luciferase reporter fused to either an ERE (A) or a CRE (B), or with a luciferase reporter lacking a specific response element (C), and with CREB1 expression vectors as indicated. Cells were stimulated with vehicle alone (DMSO, indicated with a C), E2, or FI as indicated in the Materials and Methods section. **(D)** Effect of transient siRNA-mediated knockdown of CREB1 on ERα and CREB1 activities. Luciferase reporter assays with transfected MDA-MB-134 cells, treated as indicated. All values marked with an asterisk are statistically significantly different from their respective vector or siRNA controls with P-values < 0.05; those marked with a small circle are significantly different from the S133D sample.

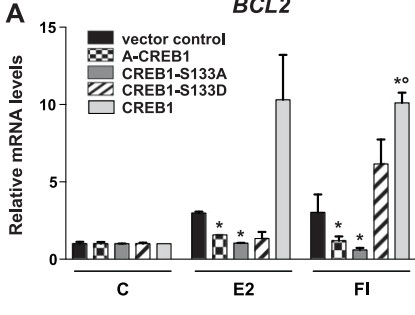

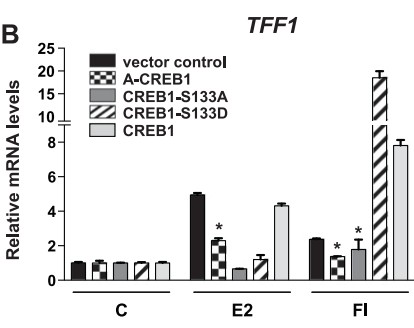

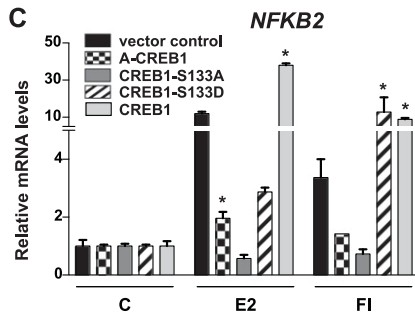

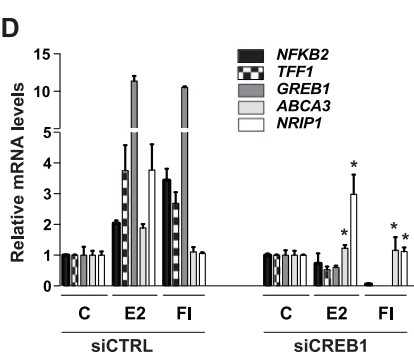

**Figure 2. CREB1 promotes the induction of endogenous ERα target genes.**
**(A–C)** CREB1 overexpression promotes the expression of the indicated ERα target genes. MDA-MB-134 cells were transfected with the indicated expression vectors and stimulated with vehicle, E2, or FI for 6 h before extraction of RNA for quantitative RT-PCR analysis of the indicated ERα target genes. See the Materials and Methods section for more experimental details. All values marked with an asterisk are statistically significantly different from their respective vector or siRNA controls with P-values ≤ 0.05; those marked with a small circle are significantly different from the S133D sample.
**(D)** Effect of transient siRNA-mediated knockdown of CREB1 on the expression of a panel of ERα target genes, as determined by quantitative RT-PCR analysis; cells and treatments as in panels A–C. All values marked with an asterisk are statistically significantly different from their respective siRNA controls with P-values < 0.05.

We then aimed to determine which ERα domains functionally collaborate with CREB1. For this purpose, we used a series of vectors expressing different ERα domains fused to the heterologous Gal4 DNA-binding domain (Fig S3). To avoid interferences with endogenous ERα, we switched to ERα-negative HEK293T cells. We found that only the fusion protein containing the ERα hormone-binding domain (HBD) (Fig S3E) responds to the overexpression of wild-type and mutant CREB1 similarly to wild-type ERα in our previous experiments. Full-length CREB1, as a fusion protein with the Gal4 DNA-binding domain, served as a control and responded like the endogenous CREB1 (Fig S3F).

Taken together, these results are consistent with an earlier report that had primarily focused on the synergy between E2 and cAMP (Lazennec et al, 2001); they extend it to the hypothesis that CREB1 plays a crucial role in promoting ERα activity in response to both E2 and cAMP. This appears to depend on CREB1 DNA binding and its phosphorylation on S133 by PKA and on the ERα HBD.

### CREB1 influences ERα recruitment to chromatin

The next experiments were designed to explore the underlying molecular mechanisms. We performed chromatin immunoprecipitation (ChIP) experiments with MDA-MB-134 cells to see whether CREB1 influences the recruitment of ERα to endogenous ERα target genes (Fig 3). The results are consistent with the effects on ERα-mediated target gene expression. Overexpression of wild-type CREB1 leads to a massive increase in ERα recruitment to ERα binding sites associated with the genes *TFF1*, *GREB1*, and *NFKB2*. In contrast, the overexpression of the mutants A-CREB1 or S133A essentially annihilates ERα recruitment to these same sites; this is very clear, despite some variability in ERα recruitment with the

distinct empty vector controls of these separate experiments (Fig 3A–C). The siRNA-mediated knockdown of CREB1 reduced the recruitment of ERα to those test genes (Fig 3D) whose expression is reduced by CREB1 knockdown (Fig 2D). Interestingly, the recruitment of ERα to the subset of ERα target genes, *ABCA3* and *NRIP1*, whose expression is not regulated by cAMP (Fig 2D) was not perturbed by the knockdown of CREB1. This indicates that recruitment to chromatin targets and transcriptional activation can be distinct steps, notably for the expression of some ERα target genes in response to cAMP. Similar results could be obtained with MCF7 cells; the effects of overexpression of CREB1 or CREB1 mutants and of the shRNA-mediated knockdown of CREB1 on ERα recruitment to three target genes point out a comparable requirement for ERα in this unrelated ERα-positive breast cancer cell line (Fig S4).

### ERα and CREB1 interact

Our results suggested that ERα might interact with CREB1. We could demonstrate with co-immunoprecipation experiments that pCREB1 is associated with endogenous ERα in MDA-MB-134 cells and that this interaction is stimulated by both E2 and cAMP (Fig 4A). This interaction may also involve CARM1, an ERα coactivator whose interaction with ERα is known to be stimulated by E2 and cAMP (Chen et al, 2000; Lee et al, 2005; Carascossa et al, 2010; Bennesch et al, 2016). We not only see a stimulation of the co-immunoprecipitation of ERα with CARM1 by activation of ERα, we also see a specific co-immunoprecipitation with CARM1 of pCREB1, whose total amounts are increased by treatment of the cells with FI (Fig 4B and C). Essentially the same stimulated interaction of ERα with CREB1 is observed with exogenously expressed HA-tagged CREB1 (Fig 4D), but intriguingly, the interaction with

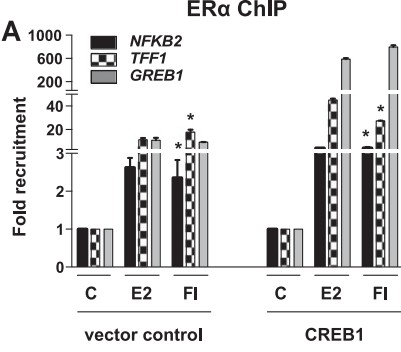

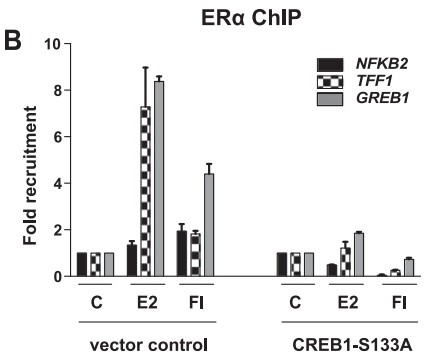

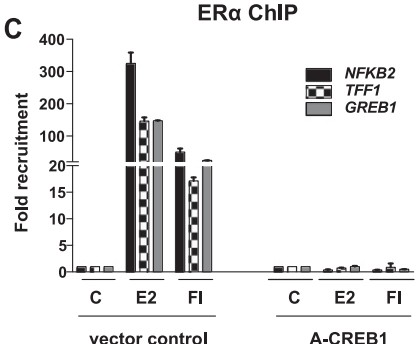

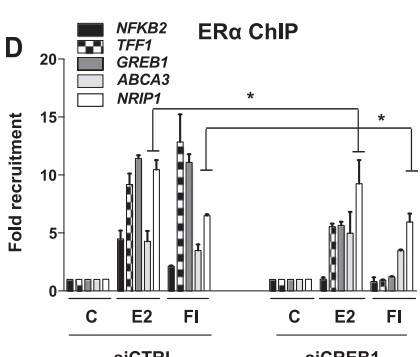

**Figure 3. CREB1 promotes the recruitment of ERα to some target genes.**
**(A–C)** ChIP-qPCR assays to determine the importance of CREB1 and its phosphorylation on S133 for the recruitment of ERα to target genes. MDA-MB-134 cells were transfected and treated as indicated in the figure and in the Materials and Methods section. Note that the fold recruitment values for ERα are experimentally variable between different experiments; these values should only be compared between different treatments and conditions within the same experiment/panel. **(D)** ChIP-qPCR assays to determine the impact of a siRNA-mediated knockdown of CREB1 on the recruitment of ERα to a wider panel of target genes. CREB1 was knocked down in MDA-MB-134 cells before stimulation as indicated. All values marked with an asterisk are statistically significantly different from their respective vector or siRNA controls with *P*-values < 0.05.

CREB1 S133A can only be revealed upon treatment with E2 (Fig 4E). In contrast, the S133D mutant displays a weak constitutive interaction, which can be further boosted with E2 (Fig 4F). We conclude from these experiments that ERα and CREB1 are part of a complex, which is likely to include yet other factors such as CARM1 (see also below) and whose assembly is stimulated by the activation of ERα and promoted by phosphorylation of CREB1 in signal-specific ways.

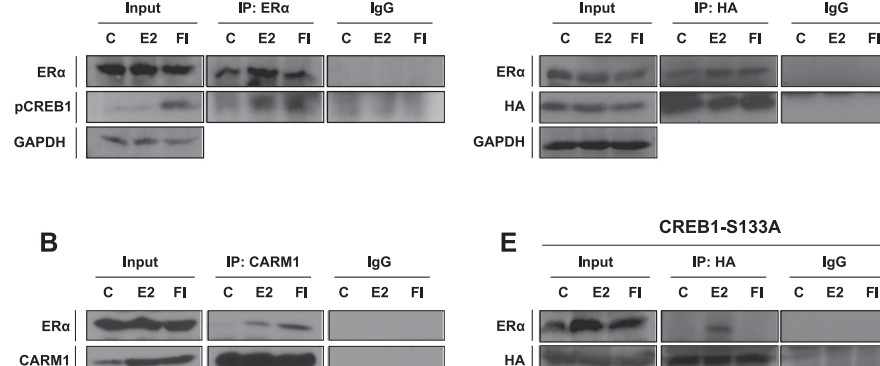

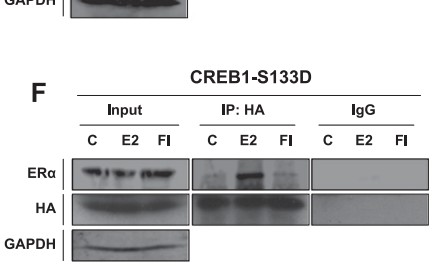

**Figure 4. ERα, CREB1, and CARM1 interact in a stimulus-dependent manner.**
**(A–C)** Co-immunoprecipitation of endogenous ERα, CARM1, and pCREB1 as a function of stimulus. MDA-MB134 cells were stimulated with DMSO (indicated by a C), E2, or FI before cell lysis and immunoprecipitation as indicated in the figure and in the Materials and Methods section. Immunoprecipitations were performed with the antibodies indicated above the immunoblots, that is, either antibodies against ERα or CARM1 or a corresponding control antibody (IgG). Proteins revealed by immunoblotting are indicated on the left. **(D–F)** ERα as a function of stimulus. Exogenous HA-tagged wild-type and mutant CREB1 were immunoprecipitated from extracts of transfected MDA-MB-134 cells to assess the association with endogenous ERα by immunoblotting with an antibody against the HA tag or ERα.

## ERα/pCREB1 crosstalk at the genome-wide level

To explore the synergy between ERα and pCREB1 at the genome-wide level, we performed ChIP-seq experiments with MDA-MB-134 cells (Fig 5). As expected, the binding of both transcription factors significantly changed upon either E2 or cAMP stimulation (Fig 5). This is most striking for pCREB1, which exhibits a major increase once cells are stimulated with cAMP (>3,500 events), compared with E2 (<500) (Fig 5B, D, and F). Opposite effects were found for ERα, where the E2 response leads the receptor to occupy more than 20,000 binding sites (Fig 5A, C, and E). pCREB1 is recruited to more than 2,000 ERα-binding sites of which the majority is induced by FI treatment (Fig 6A). Indeed, more than half (55%) of the pCREB1 target sites stimulated by only cAMP are shared with ERα binding events (Fig 6B). Unlike FI, E2 treatment prompted pCREB1 to occupy only a small part (0.6%) of the huge number of stimulated ERα target sites, even though 50% of those events are shared between the two transcription factors (Fig 6C). An indication of the functional context of the ERα or pCREB1 binding sites could be obtained by determining the genomic distribution of their binding sites (Figs S5 and 6D). In the absence of stimulation, the binding sites for pCREB1 were significantly enriched at promoters and distant intergenic regions, of which many could correspond to enhancers, and in 5′ UTR regions. The E2- or cAMP-induced sites did not show the apparent promoter and/or 5′ UTR enrichment and retained a genomic distribution that was similar to that of ERα. The genomic distribution of the shared binding sites also showed a strong enrichment at promoters and at distant intergenic regions upon E2 stimulation whereas that of the ligand-independent binding sites was similar to the non-induced ones (Fig 6D).

In view of validating the ChIP-seq data for some selected genes, which are associated with coinciding ERα and pCREB1 peaks, we extracted the data for the three ERα target genes *NFKB2*, *TFF1*, and *GREB1* (Fig 6E). These data could be confirmed by targeted ChIP-

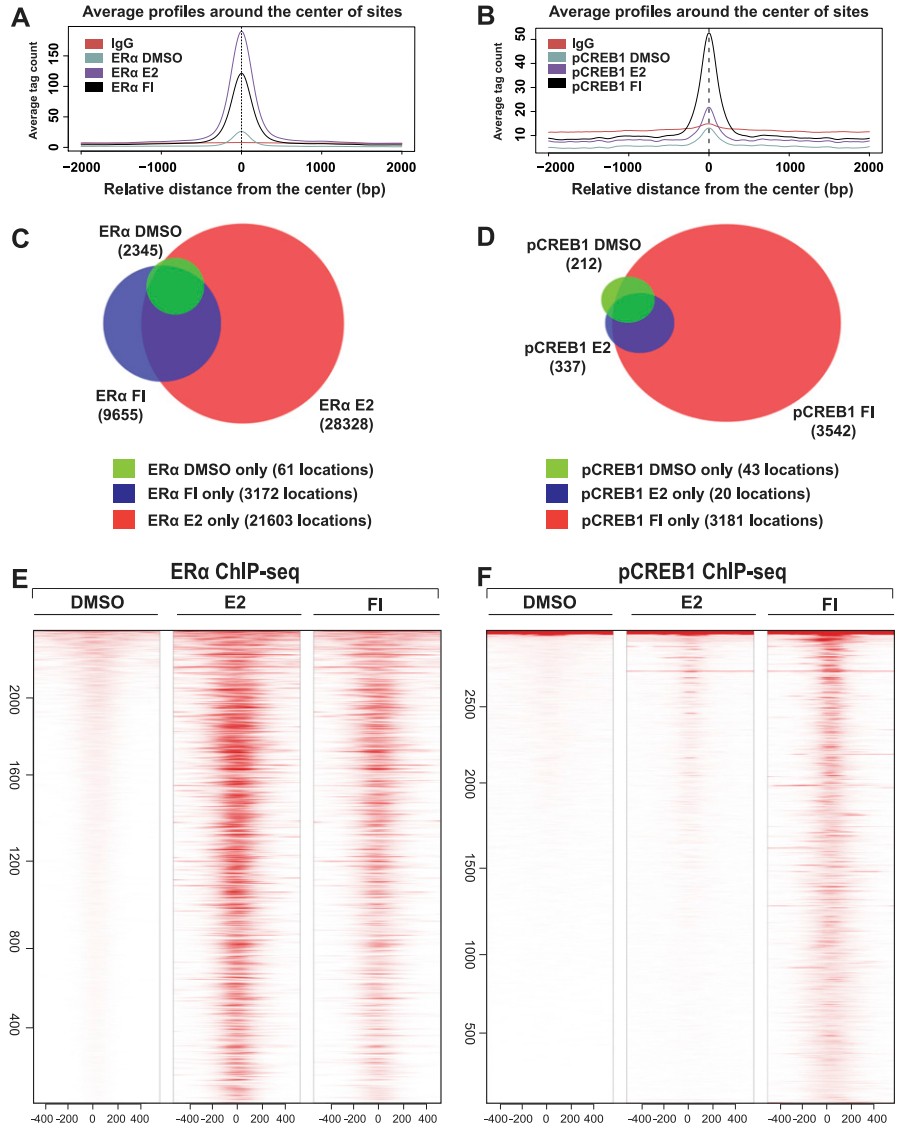

**Figure 5. Genome-wide mapping of ERα and pCREB1 chromatin binding sites in response to E2 and cAMP.**
**(A, B)** Average signals for the ChIP-seq datasets for ERα (A) and pCREB1 (B), obtained with MDA-MB-231 cells treated as indicated. The peak intensities are centered at the transcription factor peaks with a 2-kb window. **(C, D)** Venn diagrams of ERα (C) and pCREB1 (D) chromatin-binding sites as a function of treatment. Below the Venn diagrams, the number of sites (locations) that do not overlap between treatments are indicated. **(E, F)** Heat maps showing the ChIP-seq signals of ERα (E) and pCREB1 (F) in response to the indicated stimuli. Chromatin binding sites are sorted according to decreasing ChIP signals and centered at the respective transcription factor peaks with a 500-bp window on either side.

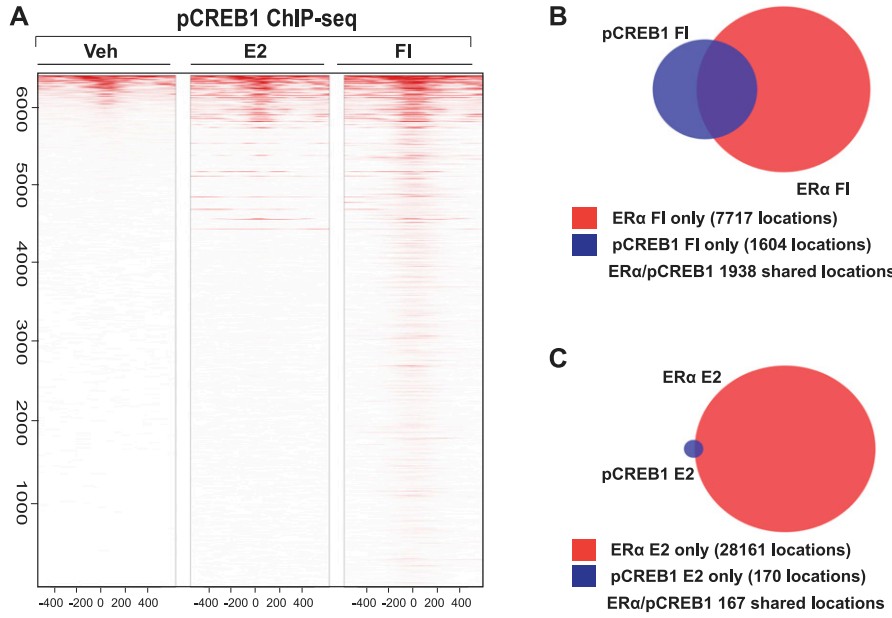

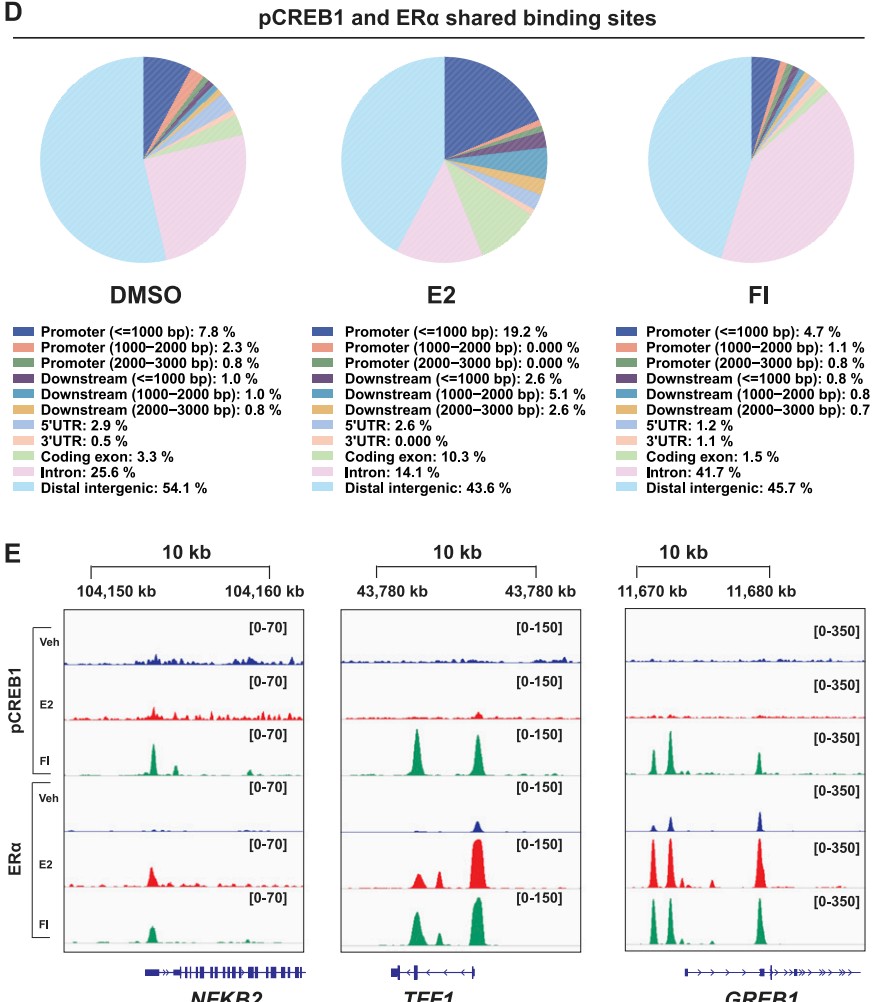

**Figure 6. Chromatin binding sites shared by pCREB1 and ERα are predominantly cAMP induced.**
**(A)** Heat maps showing the ChIP-seq signals of pCREB1 over the ERα binding sites in response to the indicated stimuli. Regions are sorted according to decreasing signals in pCREB1 binding. Data are centered at the ERα peaks with a 500-bp window on either side. All data are from MDA-MB-134 cells. **(B, C)** Venn diagrams of the cAMP-induced sites (B) and the E2-induced sites (C) indicating those shared between ERα (red) and pCREB1 (blue). **(D)** Genomic distribution of the peaks shared between ERα and pCREB1 in response to the indicated treatments. **(E)** Snapshots from the ChIP-seq data for both ERα and pCREB1 peaks are shown across indicated genes for DMSO (blue), E2 (red), and FI (green). Genomic coordinates are indicated. The numbers in square brackets in each panel indicate the scale based on the numbers of reads.

qPCR (Figs S6 and S7) and extended by monitoring the effects of overexpressing A-CREB1 or wild-type CREB1. For the same target sites and a few more shared ones, A-CREB and CREB1 clearly blocked and augmented pCREB1 recruitment, respectively (Fig S6).

In addition, we selected several genes that may be exclusively regulated by one or the other transcription factor based on our ChIP-seq data (Figs S7 and S8). For the genes *ADCY9*, *RET*, and *ARHGEF26*, which only showed sharp peaks for ERα, we could not find any binding events of pCREB1, irrespective of treatment, within a 20–60-kb window around the promoter region and/or gene. Complementary results were obtained by exploring the unique binding events of activated pCREB1: for the genes *MRPL55*, *CCDC59*, and *RBM42*, and unlike for the ERα target gene *GREB1* included as a positive control, we could not find any ERα binding event within 50 kb of the promoter region (Figs S7 and S8). As could be predicted, the knockdown of CREB1 did not impair ERα binding to the unique ERα binding sites (Fig S7E).

Thus, it appears that ERα and CREB1 may primarily collaborate when their DNA-binding sites overlap or are nearby. Our immunoprecipitation experiments showed that the two transcription factors are part of a complex. Although we cannot exclude that they interact directly, it is likely that common coregulators with different interaction surfaces for the two might contribute to bringing them together. An obvious candidate for this is CBP/p300 (Kwok et al, 1994; Chakravarti et al, 1996; Webb et al, 1998). Indeed, when we overexpressed CBP, ERα recruitment was dramatically stimulated and further boosted by the co-overexpression of CREB1; the overexpression of the CREB1 mutant S133A abolished the stimulation by CBP (Fig S9).

The gene-specific regulatory environment may be a key parameter in determining whether ERα and CREB1 coordinately affect expression. Our transfected ERE-Luc reporter (Figs 1 and S2) may be a special case because ERα responses are clearly CREB1 dependent despite the apparent absence of a CREB1 response element. Although it lacks a canonical CREB1 binding site, notably a promoter-proximal one, it contains many fortuitous CREB1 half sites throughout the plasmid. It is known that CREB1 half sites are in many instances sufficient for binding and transcriptional regulation (Mayr & Montminy, 2001; Zhang et al, 2005). The particular topology of this transiently transfected and circular reporter gene may favor the assembly of a CBP/p300-stabilized ERα/CEBP1 complex as if it were controlled by a shared binding site.

Collectively, the genome-wide analysis provides a view of the complexity of the ERα/pCREB1 crosstalk. Most chromatin binding events shared between these factors are cAMP responsive, suggesting that at the genome-wide level pCREB1 assists ERα activity principally in the absence of its cognate ligand and for a limited number of ERα target genes. Our results with CBP overexpression suggest that some of this synergy may be orchestrated by common coregulators.

### ERα and CREB1 contribute to protect breast cancer cells against apoptosis

ERα is known to exert a protective role against apoptosis, in part by inducing the expression of anti-apoptotic genes such as *BCL2* (Eguchi et al, 2000; Cericatto et al, 2005; Bratton et al, 2010). CREB1

plays a pro-survival role against induced death signals (Wilson et al, 1996; Finkbeiner, 2000; Shankar et al, 2005; Shukla et al, 2009; Schoknecht et al, 2017; Shabestari et al, 2017). Thus, ERα and CREB1 might collaborate to protect cells against apoptosis. To investigate this, we performed knockdowns of either ERα or CREB1 in MDA-MB-134 cells (Fig S1B and C) and treated the cells with either E2 or cAMP for 24 h before subjecting them to an apoptotic stimulus. After the treatment with staurosporine (STS), we observed that the mitochondrial membrane potential is reduced, a marker of mitochondria-mediated cell death. We found that E2 and cAMP reduced mitochondrial depolarization (Fig S10A). Knocking down ERα or CREB1 suppressed this protective effect (Fig S10B and C), suggesting that the latter is mediated by ERα and CREB1. As a more direct assessment of apoptosis, we examined the effects of various treatments by determining the changes in nuclear morphology. This analysis confirmed what we had seen by looking at the mitochondrial membrane potential, which is a protective effect of ERα- and CREB1-mediated responses to E2 and cAMP (Fig 7A).

### *CREB1* expression levels correlate with outcome in breast cancer

To determine whether the expression level of *CREB1* has any clinical significance, we mined outcome-linked gene expression data for breast cancer using the "Gene Expression-Based Outcome for Breast Cancer Online" (GOBO) tool (Ringnér et al, 2011) and displayed the outcomes in Kaplan–Meier curves. If one considers all types of ERα-positive breast tumors, higher levels of *CREB1* expression are significantly associated with better outcome, in this case improved "distant metastasis-free survival" (Fig 7B). For ERα-positive tumors, this benefit of higher *CREB1* levels disappears in the subset that is lymph node positive, and thus potentially more aggressive, and in those that are being treated with the anti-estrogen tamoxifen, the standard endocrine therapy (Fig S11). The prognosis for ERα-negative breast tumors is not affected by CREB1 levels in this dataset (Fig 7B). Although the GOBO database has a large number of breast cancer samples (1,881), we decided to confirm these results by mining another database. Using the "Kaplan–Meier Plotter" (Lanczky et al, 2016), we interrogated "The Cancer Genome Atlas" (https://cancergenome.nih.gov). Although the numbers of samples for certain categories and analyses were lower than that with GOBO, we found essentially the same results when we analyzed those that were comparable with both tools (Fig S12). Overall, these findings are consistent with the notion that a functional ERα/CREB1 interplay is a positive predictor in breast cancer.

## Discussion

Our study confirms and substantially complements the initial report (Lazennec et al, 2001) on a functional interaction between ERα and CREB1. Lazennec et al (2001) found that CREB1 stimulates ERα activity on a transfected reporter gene and that this can be blunted with a dominant-negative mutant of CREB1. Furthermore, they demonstrated an interaction with a mammalian two-hybrid assay. Here, we report that CREB1 and ERα can be biochemically shown to be part of the same complex and that CREB1 is required for ERα recruitment and responses. Our genome-wide analysis highlights

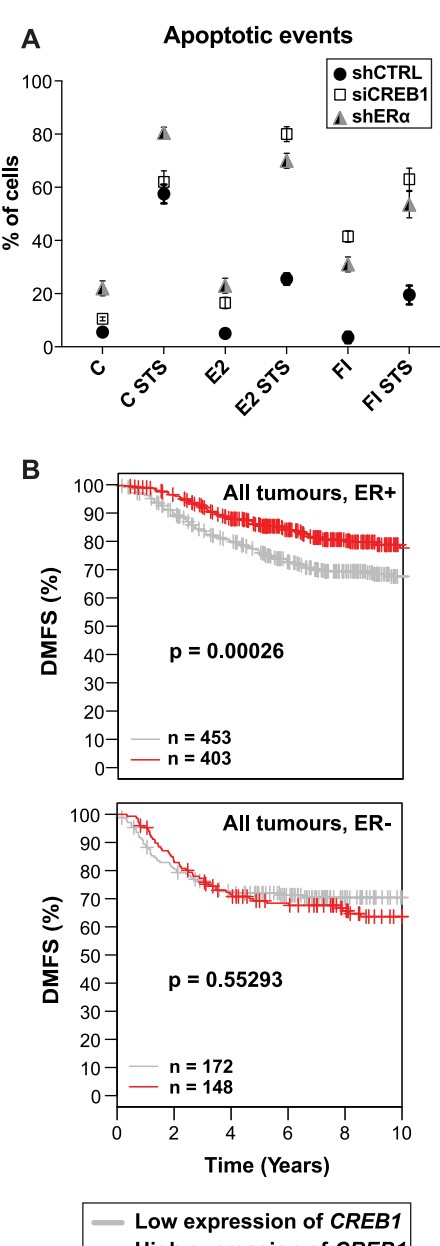

**A**

**Apoptotic events**

Legend:
- ● shCTRL
- □ siCREB1
- ▲ shERα

y-axis: % of cells (0 to 100)
x-axis: C, C STS, E2, E2 STS, FI, FI STS

**B**

All tumours, ER+

DMFS (%) vs Time (Years)

p = 0.00026

n = 453
n = 403

All tumours, ER-

DMFS (%)

p = 0.55293

n = 172
n = 148

— Low expression of *CREB1*
— High expression of *CREB1*

**Figure 7. CREB1 and ERα differentially protect breast cancer cells and patients.** **(A)** Both CREB1 and ERα suppress STS-induced apoptosis in response to E2 and cAMP. Knockdown of CREB1 and ERα, treatments with STS, E2, and FI, and microscopic assessment of apoptosis were performed with MDA-MB-134 cells as described in the Materials and Methods section. The data points are averages of three independent experiments obtained by the inspection of 200 cells each, and the error bars indicate the standard error of the mean. **(B)** High expression levels of the *CREB1* gene correlate with better breast cancer outcome. The *CREB1* gene was used as a marker for the Kaplan–Meier analysis of breast cancers with the GOBO tool. The plots indicate distant metastasis-free survival (DMSF) as a function of time for patients with ERα-positive (ER+) (upper panel) and ERα-negative (ER–) (lower panel) breast cancers. Within each panel, the number of samples in each category and the *P*-value for the difference between high and low expression cases are shown. Additional analyses are presented in Figs S11 and S12.

the complex crosstalk between these two transcription factors and sets the stage for a better understanding of the biological synergy that may underlie the protective effects of both E2 and cAMP

signaling against apoptosis and of high CREB1 levels in breast cancer patients.

### CREB1 stimulates and is required for ERα activity

Our results demonstrate that CREB1 in its activated phosphorylated form stimulates ERα activity. CREB1 supports both the E2-activated liganded form and the cAMP-activated unliganded form of ERα. Defective CREB1 mutants, including one that cannot be phosphorylated on a critical serine residue (S133A) and the CREB1 knockdown repress ERα activity. The latter finding argues that CREB1 itself is required in a non-redundant fashion despite the fact that it is only one member of a multiprotein family. Considering that CREB1 affects ERα activated either by cognate steroid ligand or through a ligand-independent pathway, we can argue that CREB1 acts downstream of the earliest steps in ERα activation, but upstream of ERα chromatin recruitment. This is supported by the ChIP and ChIP-seq experiments, which indicate that CREB1 promotes the recruitment of ERα to target sites, in particular to those that are shared between the two transcription factors. At least for these shared sites, CREB1 becomes a requirement for ERα recruitment and, as a consequence of that, of the transcriptional regulatory activity of ERα for the genes that are under the control of these sites. In fact, our results strongly argue that it is the phosphorylated version of CREB1, pCREB1, that is required. Upon cAMP signaling, the proportion of pCREB1 is expected to increase and it does (Fig 4). What exactly happens upon E2 signaling still needs to be clarified. Either the pCREB1 requirements are met by whatever basal level phosphorylated CREB1 there is, or there is a transient increase, which we have failed to see in our experiments (Fig 4), in response to the non-genomic stimulation of adenylate cyclase activity by E2 (Aronica et al, 1994; Levin & Hammes, 2016; Barton et al, 2018). Because ERα is a target of PKA as well (Anbalagan & Rowan, 2015), one must wonder whether its phosphorylation plays any role in the synergy described here. Although we have not directly addressed this issue in this context, the first report on ERα/CREB1 synergy came to the conclusion that it is not (Lazennec et al, 2001), and we found that an ERα mutant that cannot be phosphorylated on S305, the main PKA target site, is still activated by cAMP (Carascossa et al, 2010).

It is remarkable that CREB1 is also needed for the ligand-independent activation of ERα by cAMP signaling. Overall, we now know of three factors that are required and are targets of PKA. Upon phosphorylation by PKA, CARM1 can directly interact with the ERα HBD in the absence of estrogen (Carascossa et al, 2010), possibly promoting the recruitment of yet other factors. The deacetylase LSD1 may be one of them, because its phosphorylation by PKA and recruitment to ERα are necessary for ERα activity (Bennesch et al, 2016). pCREB1 is the third factor that we have found to be needed to relay the cAMP signal and to promote the activation of ERα. At this point, we do not know the exact order of events nor can we exclude that the phosphorylation and recruitment of yet other factors may be needed. It should be pointed out that this mode of activation of ERα still requires an active AF2 and presumably many of the factors that associate with it (Carascossa et al, 2010). The phosphorylation of multiple factors by PKA might promote the assembly of a multiprotein complex on the ERα HBD, which ultimately favors the conformational change of the HBD

converting AF2 into its active state, even in the absence of the estrogenic ligand, which normally triggers this transition.

### Molecular mechanism of transcription factor synergy

The key question is how the synergy of ERα and CREB1 comes about at the molecular level. Several possibilities are conceivable: (i) Both transcription factors contact specific response elements; for shared chromatin binding sites, they may be sufficiently close to allow the interaction between ERα and CREB1 to favor and to stabilize target site engagement; the interaction may be indirect and/or strengthened by shared coregulators such as CBP/p300, and it might occur over larger distances by looping. (ii) Only one of the two directly binds DNA whereas the other tethers to it. ERα is well known to be able to tether to AP-1, NF-κB, and CREB1 (Heldring et al, 2007, 2011). Interestingly, a DNA-binding defective CREB1 mutant prevents ERα from tethering not only onto DNA-bound CREB1 but also onto AP-1 (Heldring et al, 2011). This indicates that CREB1 is more generally required for ERα function through tethering. (iii) One acts as the pioneer factor for the other. Pioneer factors are a particular class of transcription factors, which can bind compacted chromatin. Once they have initiated chromatin remodeling, the DNA becomes more accessible for "standard" transcription factors. This concept has been extensively elaborated for ERα, for which various factors such as FoxA1 act as pioneer factors (Jozwik & Carroll, 2012). However, the distinction between pioneer factors and transcription factors does not sufficiently take into account the dynamic aspects of chromatin interactions. An alternative and complementary view posits that the boundaries between pioneer factors and transcription factors may be somewhat arbitrary (Voss et al, 2011; Swinstead et al, 2016b). Through assisted loading, one transcription factor, for example the GR, may promote the activity of another one such as ERα (Voss et al, 2011).

Several of these mechanisms may be involved in the ERα/CREB1 crosstalk, which we have described here. A sequence motif analysis for the shared binding sites supports the idea that multiple mechanisms might be at work (Fig S13). Not surprisingly, the most common sequence motifs in ERα and CREB1 binding sites, irrespective of treatment, correspond to canonical EREs and CREs (Fig S13A). Whereas a substantial proportion of the nearly 2,000 shared binding sites in response to cAMP (Fig 6B) contain EREs and/or CREs (Fig S13B), only a minority of the 167 sites in response to E2 (Fig 6C) do so (Fig S13B). Clearly, only an in-depth analysis of individual target sites will allow one to discriminate between different mechanisms. For many of the ERα target sites we have focused on, a direct contact of ERα with the DNA is known to be involved. For CREB1, direct DNA binding must be involved at least for all those cases where we showed that the overexpression of a DNA-binding defective mutant blocks ERα activity. Regarding the DNA-binding requirement, this situation is reminiscent of the E2- and TNFα-induced crosstalk between ERα and NF-κB. In this case, the concomitant activation of NF-κB redirects ERα to shared sites (Franco et al, 2015). At a genome-wide level, discriminating between different modes of action, as outlined above, may be somewhat semantic because the assembly of large regulatory structures, such as those previously termed superenhancers or MegaTrans enhancers (Hnisz et al, 2013; Liu et al, 2014), provides a complex framework for concomitant yet dynamic interactions between multiple transcription factors, coregulators, DNA, and chromatin components.

### Physiological relevance of ERα/CREB1 synergy

ERα and CREB1 have long been known, separately, to be protective factors against apoptosis (Wilson et al, 1996; Eguchi et al, 2000; Finkbeiner, 2000; Cericatto et al, 2005; Shankar et al, 2005; Shukla et al, 2009; Bratton et al, 2010; Schoknecht et al, 2017; Shabestari et al, 2017). This correlates with their ability to induce directly the expression of anti-apoptotic genes such as *BCL2*. We have confirmed with MDA-MB-134 breast cancer cells that both E2 and cAMP signaling protect cells against an apoptotic stimulus. The protective effect of these signals is mediated by ERα and CREB1 because their knockdown sensitizes cells (Figs 7A and S10). The *BCL2* gene is likely to be an important target, but considering the large size of the ERα and CREB1 cistromes, including the shared sites, there may be many more relevant target genes. To gain a better understanding of how ERα and CREB1 act to protect cells, these genes will have to be identified. Moreover, their in-depth analysis should allow one to determine whether they are regulated independently by ERα and CREB1 or whether they are subject to the synergistic regulation described here.

Our analysis of publicly available gene expression profiles revealed that high levels of CREB1 are associated with better prognosis in ERα-positive breast cancers, i.e., that they must have a protective effect in patients as well, and not just in cells. At first, it may seem paradoxical that protecting breast cancer cells against apoptosis at the cellular level may translate into a beneficial effect in patients. However, this might be resolved by the fact that a functional ERα is associated with better prognosis in breast cancer. Being required for ERα function, high levels of CREB1 may contribute to prevent a tumorigenic evolution, notably for cancers that have not yet been treated with tamoxifen, toward more aggressive and therapy-resistant versions of breast cancer that are estrogen and ERα independent.

# Materials and Methods

### Antibodies and reagents

The rabbit polyclonal antisera HC-20 and A300-498-A against ERα were purchased from Santa Cruz Biotechnology and Bethyl Laboratories, respectively; the rabbit polyclonal antiserum A300-421A against CARM1 was purchased from Bethyl Laboratories; the rabbit monoclonal antibodies against pCREB1 (87G3) and CREB1 (48H2) were purchased from Cell Signaling Technology; the mouse monoclonal antibody against the tag HA.11 (clone 16B12) was purchased from BioLegend; the mouse monoclonal antibodies against the FLAG tag (clone M2) and α-tubulin (T9026) were purchased from Sigma-Aldrich; the mouse monoclonal antibody against GAPDH (6C5) was purchased from Abcam. Corresponding control IgGs from rabbit (Cat. No. I5006) and mouse (I5381) were purchased from Sigma-Aldrich.

17β-estradiol (E2) was purchased from Sigma-Aldrich, forskolin and STS from Chemie-Brunschwig, and 3-isobutyl-1-methylxanthine from Millipore. All reagents were dissolved in DMSO as vehicle.

### Plasmids

The empty expression vector pCMV4 was from Addgene. The following series of expression vectors based on plasmid pCMV5-HA were obtained through the "MRC PPU Reagents and Services Facility" at the University of Dundee: pCMV5-HA CREB1 (# DU4071), pCMV5-HA CREB1 S133A (# DU4073), and pCMV5-HA CREB1 S133D (# DU4106). The expression vectors pCMV-FLAG A-CREB1 (Ahn et al, 1998), pRC/RSV-mCBP-HA (Chrivia et al, 1993), and HEG0 for wild-type ERα (Tora et al, 1989) have been described.

For luciferase reporter assays, the following plasmids were used: the luciferase reporters XETL (here referred to as ERE-Luc) for ERα (Bunone et al, 1996), CRE-Luc for CREB1 (Stratagene), XTL (here referred to as Luc) without response element (Bunone et al, 1996) as a negative control, and plasmid GK1 for the Gal4 fusions (Webb et al, 1998). The Renilla luciferase pRL-CMV (Promega) was used as transfection control. The expression vector series pSCTEV gal93 was used for expression of the Gal4 DNA-binding domain (aa 1–93) fusions (Seipel et al, 1992) with the ERα HBD and AF2 (aa 282–595) (Maggiolini et al, 2001), the N-terminal domain of ERα with the activation function AF1 (aa 82–152) (Gburcik et al, 2005), the ERα DNA-binding domain (aa 180–262) (Carascossa et al, 2010), and the ERα hinge region (aa 251–300) (Carascossa et al, 2010). The expression vector for Gal4 fusion with full-length CREB1 was a gift from Ugo Moens (Delghandi et al, 2005).

For the knockdown of ERα, cells were infected with pLKO.1-based lentiviruses expressing either a specific shRNA against ERα (from construct pLKO.shESR1) (Bennesch et al, 2016) or against CREB1 (Table S1) or scrambled shRNA as control (from construct pLKO.shCTRL) (Sarbassov et al, 2005). The plasmids psPAX2 and pMD2G (gift from Didier Trono's laboratory at EPFL) were used for preparing the lentiviruses.

### Cell culture and transfection experiments

HEK293T, MDA-MB-134, and MCF7 cells were maintained in Dulbecco's modified Eagle's medium supplemented with 10% FBS and 1% penicillin-streptomycin (P/S). Before transfection experiments, cells were cultured for 3 d in phenol red–free medium, supplemented with 5% charcoal-stripped FBS (5% ch-FBS), 1% P/S, and 1% L-glutamine. Except for siRNA experiments, transient transfections of different plasmid constructs were performed with polyethylenimine MAX 4000 (Polysciences) for 24 h in serum- and P/S-free medium supplemented with glutamine. Subsequently, after replacing this medium with fresh phenol red– and serum-free medium, the cells were treated with the vehicle DMSO, 100 nM E2, or the cocktail FI (consisting of 10 μM forskolin and 100 μM 3-isobutyl-1-methylxanthine) for 1, 6, and 24 h before harvesting for ChIP and immunoprecipitation, RT-PCR, and luciferase assays, respectively.

### shRNA- and siRNA-mediated knockdowns

For the production of lentiviruses, HEK293T cells were seeded to a 60% confluency in medium 24 h before polyethylenimine transfection in complete medium with plasmids pLKO.shESR1, pLKO.shCREB1, or pLKO.shCTRL plus plasmids pMD2G and psPAX2. 24 h

later, the medium was replaced with fresh complete medium and lentivirus-containing supernatants were collected every 12 h during 48 h. These supernatants were used to infect MDA-MB-134 or MCF7 cells during 72 h. After infection, cells were subjected to selection with 3 μg/ml puromycin for at least 24 h to eliminate any remaining non-infected cells. After selection, cells were collected for experiments or maintained in normal culture medium, with 1 μg/ml of puromycin, for not more than a week. The efficiency of the knockdown was tested each time by immunoblotting. For siCREB1-mediated knockdowns, MDA-MB-134 cells were transfected with a final concentration of 20 nM siRNA, purchased as a pool of four individual siRNAs targeting CREB1 (ON-TARGETplus Human CREB1 siRNA) and a pool of four siRNAs designed as negative control (ON-TARGETplus Non-targeting Pool siRNA) from Dharmacon. Where indicated, siRNAs were cotransfected with plasmid DNAs; transfections were performed using Jet-Prime (PolyPlus) in medium without serum and P/S, supplemented with glutamine for 4 h. After infection and selection, or siRNA/plasmid transfection, the medium was replaced with phenol red– and serum-free medium, and the cells were incubated for 24 h before performing experiments (such as inducing cells for 24 h for luciferase or apoptosis assays).

### Immunoprecipitation experiments

Cells were washed once with ice-cold PBS and lysed with a buffer containing 10 mM Tris–HCl, pH 7.5, 50 mM NaCl, 1 mM EDTA, 1 mM DTT, 10% glycerol, 10 mM Na-molybdate, and a protease inhibitor cocktail (Thermo Fisher Scientific). Cells were broken by sonication with 40 on/off cycles of 30 s at high power using the Bioruptor sonicator (Diagenode). After that, cell lysates were centrifuged at maximal speed at 4°C for 10 min. Supernatants were recovered whereas pellets containing the cell debris were discarded. Protein concentrations of lysates were measured using the Bradford protein assay. 100 μg proteins were saved for input and 2 mg were incubated with protein G–dynabeads (Life Technologies) precoated with specific antibody or with control antibody of the same species by overnight rotation at 4°C. Beads were washed 4×, 10 min each, with lysis buffer containing 0.1% Triton X-100, and then boiled in sample buffer containing 100 mM DTT.

### Extraction of total cellular protein for immunoblotting

Cells were washed once and pelleted with ice-cold Tris-buffered saline. Cells were lysed in cold RIPA lysis buffer (50 mM Tris–HCl, pH 7.4, 150 mM NaCl, 1% Triton X-100, 10% glycerol, 1 mM EDTA, and 1 mM DTT) supplemented with a protease inhibitor cocktail (Thermo Fisher Scientific), and sonicated 15× for 30 s at high power using the Bioruptor sonicator (Diagenode). Lysates were centrifuged at maximum speed, at 4°C, and for 30 min. Supernatants were recovered and cell debris was discarded. Protein concentrations of lysates were measured using the Bradford protein assay.

### Luciferase activity assays

Firefly luciferase and Renilla luciferase control activities were measured using the Dual-Luciferase Reporter Assay System (Promega), following the manufacturer's instructions. For each

**Life Science Alliance**

condition, firefly luminescence is standardized to the activity of the Renilla luciferase transfection control. Values are expressed as fold increase relative to the DMSO-treatment conditions, arbitrarily set to 1.

### Gene expression analyses by quantitative RT-PCR

RNA extraction for RT-PCR was performed with MDA-MB-134 cells following the previously described protocol (Bennesch et al, 2016). The specific primers are listed in Table S2. mRNA levels were standardized to the mRNA of the internal housekeeping gene *GAPDH*, and expressed as fold increase relative to the DMSO-treatment conditions, arbitrarily set to 1.

### ChIP-qPCR

Cross-linking and lysis of cells for ChIP were performed following a previously published protocol (Schmidt et al, 2009). Recruitment of ERα or pCREB1 was determined by using qPCR, using specific primers and primers to a region of the *FGFR9* gene as negative control. Primer sequences are listed in Table S3. Values of each target region were normalized to their corresponding input values and then to the values of the negative control region. Recruitment is presented as the fold recruitment relative to DMSO-treated cells.

### ChIP-seq

ChIP-seq samples were generated from two biological replicates for each condition with the rabbit polyclonal antiserum A300-498-A against ERα and the rabbit monoclonal 87G3 against pCREB1. A total of 5 ng of recovered DNA was used to generate libraries for sequencing, following a previously published protocol (Schmidt et al, 2009). Sequencing adaptors were attached applying the Illumina TruSeq protocol and sequenced using an Illumina HiSeq 2500 sequencer. Analyses were done with tools of the Galaxy suite (https://usegalaxy.org). ChIP-seq libraries were aligned to the reference genome hg19 using default parameters of Bowtie 2 alignment (version 2.2.6). The enriched regions of the genome were identified by comparing sequences from treated samples to those of the IgG control sample using the MACS peak caller (version 2.1.1.20160309.0). Venn diagram (version 1.0.0), HeatMap (version 1.0.0), SitePro aggregation plots (version 1.0.0), and "Enrichment on chromosome and annotation" (CEAS, version 1.0.0) analyses were performed using the online Galaxy/Cistrome tool. Examples of shared chromatin interactions of pCREB1 and ERα were displayed using the Integrative Genomic Viewer (IGV_version 2.3.97). The criterion for calling peaks "shared" was a minimal overlap of the reads by at least 1 bp in both replicates. Motif analyses were performed using the MEME (version 4.11.1.0) and FIMO (version 4.11.10) (Grant et al, 2011) tools of the MEME suite (http://meme-suite.org) (Bailey & Elkan, 1994).

### Assay of mitochondrial membrane potential

MDA-MB-134 cells, cultured in preparation of the experiment as indicated above, were stimulated with E2, FI, or DMSO for 24 h and apoptosis was triggered with 1 µM STS during the last 4 h of each treatment. Cells were washed twice with PBS, detached by trypsinization, harvested, resuspended in 200 µl PBS, and transferred to a 5-ml FACS tube. 200 nM MitoTracker Red (Thermo Fisher Scientific) was added and the assay tubes were incubated at 37°C for 30 min. Before analyses by flow cytometry, another 200 µl PBS was added to each tube. The analyses of a set number of cells (10,000) were performed with a FACSCalibur cell analyzer (BD Biosciences) and the software CellQuest Pro; the acquired data were processed with the software FlowJo.

### Fluorescence microscopy for apoptotic events

MDA-MB-134 cells were cultured and treated as mentioned above, and then fixed in 4% wt/vol paraformaldehyde for 10 min and washed in ice-cold PBS. PBS containing 0.5% wt/vol bovine serum albumin and 0.01% of Triton X-100 was added for 30 min to permeabilize cell membranes. Cells were washed twice with cold PBS, gently rocking, and the blocking solution containing 5% wt/vol bovine serum albumin in PBS was added for 1 h at room temperature. Cell nuclei were stained with a solution containing DAPI in PBS for 30 min. Coverslips were mounted face down and pictures were acquired by microscopy using a ZEISS Axiophot. Apoptotic cells were identified based on their nuclear morphology as described (Cummings et al, 2012);. Briefly, cells characterized by a strong nuclear condensation or with chromatin clustered at the edge or periphery of the nuclear membrane were classified as apoptotic cells. For each condition, more than 200 cells were counted and the number of apoptotic cells was expressed as a percentage of the total.

### Kaplan–Meier curves

Kaplan–Meier curves (Dinse & Lagakos, 1982) were generated using the online tools GOBO (http://co.bmc.lu.se/gobo) (Ringnér et al, 2011) and "Kaplan–Meier Plotter" (http://kmplot.com/analysis) (Lanczky et al, 2016) with *CREB1* as the query gene (Entrez gene ID: 1385 and gene ID: 22572_at, respectively). Note that "Kaplan–Meier Plotter" can interrogate several databases, including The Cancer Genome Atlas.

### Statistical analyses

Unless otherwise indicated, the data shown are representative of three independent biological experiments with triplicate samples in the case of luciferase assays and technical replicates for ChIP-qPCR and RT-PCR analyses, with error bars indicating the standard deviation of the mean. For some relevant comparisons, *t* tests were performed.

### Data availability

ChIP-seq data have been deposited in the GEO repository under the series record GSE109103.

# Supplementary Information

# Acknowledgements

We thank the genomics platform of iGE3 for sequencing of the ChIP-seq samples of ERα and pCREB1, Federico Miozzo and Grégory Ségala for advice and help with the online Galaxy platform, and Suzan Stelloo for initial training for ChIP experiments. We are also grateful to Grégory Ségala for critical reading of the manuscript. This work was supported by the Swiss National Science Foundation, the Medic foundation, and the Canton de Genève.

## Author Contributions

M Berto: conceptualization, formal analysis, investigation, and writing—original draft, review, and editing.
V Jean: writing—review and editing and performed preliminary experiments.
W Zwart: conceptualization and writing—review and editing, and analysis of preliminary results.
D Picard: conceptualization, formal analysis, supervision, funding acquisition, writing—original draft, project administration, and writing—review and editing.

## Conflict of Interest Statement

The authors declare that they have no conflict of interest.

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
