## [Reviewer comments · Life Science Alliance]

ER α activity depends on interaction and target site corecruitment with phosphorylated CREB1

Melissa Berto, Valerie Jean, Wilbert Zwart, Didier Picard

DOI: 10.26508/lsa.201800055

Review timeline:

Submission date:	8 March 2018
1 st Editorial Decision:	3 April 2018
Appeal received:	16 April 2018
2 nd Editorial Decision:	18 April 2018
1 st Revision Received:	2 May 2018
3 rd Editorial Decision:	18 May 2018
2 nd Revision Received:	18 May 2018
Accepted:	22 May 2018

Report:

(Note: Letters and reports are not edited. The original formatting of letters and referee reports may not be reflected in this compilation.)

1st Editorial Decision

3 April 2018

Thank you for submitting your manuscript entitled "ER α activity depends on interaction and target site corecruitment with phosphorylated CREB1" to Life Science Alliance. The manuscript has been evaluated by expert reviewers, whose reports are appended below. Unfortunately, after an assessment of the reviewer feedback by the academic editor and myself, our editorial decision is against publication in Life Science Alliance at this stage.

As you will see, the two reviewers who evaluated the study point out several issues and think that your conclusions are not sufficiently supported by the data provided. Importantly, they note a lack of generalisation and lack of inclusion of statistical analyses. They further think that the ChIP-seq analysis is non-conclusive and that several assays additional assay such as cell proliferation assays, inclusion of TCGA database analysis, and further proof of protection from apoptosis via CREB1 would be needed.

Although your manuscript is interesting, we feel that the outcome of addressing these points is rather unclear at this stage, and we have thus decided to return your manuscript to you. We are sorry our decision is not more positive, but hope that you find the reviews constructive. Of course, this decision does not imply any lack of interest in your work and we look forward to future submissions from your lab.

Thank you for thinking of Life Science Alliance as an appropriate place to publish your work.

REFEREE REPORTS

Reviewer #1 (Comments to the Authors (Required)):

In this MS titled "ER α activity depends on interaction and target site corecruitment with phosphorylated CREB1", the authors Melissa Berto and colleagues found that CREB1 stimulation is necessary for ER α activity. CREB1 can activate some ER α target genes expression. In addition, the stimulatory activity of CREB 1 requires its DNA binding and activation by phosphorylation, and affects the chromatin recruitment of ER α . Although this finding is interesting, some of issues need to be investigated and explained.

Major Comments

- 1) In this MS, the authors only used one cell line for their study. At least ER α positive two cell lines like MCF7 and T47D, and one ER negative cell line (as a negative control) should be involved in.
- 2) Fig EV3, As we know, ER α is essential for ER α positive cell proliferation and mitochondrial membrane potential(MMP) is a key parameter of cell viability/survival, but the authors found that ER α KD increased MMP. So, cell proliferation assays should be involved in these cell lines.

Minor comments

- 1) Statistical analysis should be involved in these bar graphs,
- 2) Fig1b, the authors should explain why Vector control and CREB1S-133D group have the similar effects on LUC reporter assays.
- 3) Fig 2a, c, E2 treatment, CREB1S-133D group even lower than Control? Please explain it.
- 4) Fig4, some of the WB film are very fuzzy, Please repeat them.
- 5) TCGA database about breast cancer patients should be analyzed.

Reviewer #2 (Comments to the Authors (Required)):

The manuscript titled "ER α activity depends on interaction and target site corecruitment with phosphorylated CREB1" illuminates many novel forms of cooperativity between the estrogen signaling and cyclic AMP signaling pathways. Overall, I had a very positive impression of the manuscript, as the authors showed by multiple methodologies the ways in which ER α is dependent upon and cooperative with the cAMP signaling pathway. Starting with simple assays like luciferase assays in Figure 1, the authors demonstrated that overexpression of CREB1 increased ER α activity, whilst knockdown of CREB1 inhibited the activity of ER α to activate transcription of the luciferase construct. The authors implicated the S133 residue of CREB1 as being necessary for ER α cooperativity, since the S133A mutant is unable to be phosphorylated by the PKA enzyme and acted similarly to the dominant negative version of the CREB1 enzyme. Figures 2 and 3 deals with mRNA level and the ChIP binding levels of ER α and CREB1 proteins in MDA-MB-134 breast cancer cell line. Interestingly, many of the canonically upregulated ER α target genes in breast cancers like TFF1, GREB1 and BCL2 are also bound by ER α without any estrogen signaling, in response to Forskolin. Figure 4 shows with CoIP experiments that ER α and CREB1 interactions are stimulated by E2. Figures 5 and 6 show the genome wide distributions of ER α and S133phos CREB1, not surprisingly, many of the sites ER α is recruited to by E2 are also the same times ER α gets recruited to with treatment of Forskolin. Interestingly, this relationship is somewhat reciprocal, as roughly 60% of pCREB1 binding sites stimulated by cAMP are shared with ER α binding. Figure 7 deals with the functional importance of ER α and CREB1 in the suppression of apoptosis in the breast cancer cell model, it was already known that E2 stimulation protects cancer cells from apoptosis, this figure showed that as might be expected, this protection is also dependent on activity of CREB1. Somewhat paradoxically, this apoptotic protection is correlated with higher survival curves as higher CREB1 in ER+ is correlated with longer survival.

Major Criticisms

In Figure 3, ChIP fold recruitment of ER α to NFkB, TFF1 and GREB1 is extremely highly variable in the vector control condition, ranging from 20-fold in 3A, to 8-fold in 3B to up to 400-fold in 3C. Since these are the same cells with the same conditions, this extremely high disparity indicates that experimental different or methodological difference may be the contributing factor. With such high variation, the rest of data might be suspect.

In Figure 7A, there are no statistical significance tests. The data presented seems from be from a single experiment. Would be much stronger point with some sort of significance test.

In general, the experiments shown in this paper have no significance testing. In the first seven figures, the only p value that can be seen is in Figure 7B comparing survival curves.

Conclusion:

Overall, I found this to be an informative paper, wherein the conclusions are supported by the data. I think the authors are overselling the point about the protection from apoptosis provided by CREB1. Rather, if they should choose to make this point, they should do at least one other independent experiment to assay the antiapoptotic effects of CREB1, measurement of effects on BCL2 or the Caspase pathways.

Response to reviewers' comments regarding LSA-2018-00055-T

Importantly, they note a lack of generalisation and lack of inclusion of statistical analyses. They further think that the ChIP-seq analysis is non-conclusive and that several assays additional assay such as cell proliferation assays, inclusion of TCGA database analysis, and further proof of protection from apoptosis via CREB1 would be needed.

We are puzzled by some of these general comments.

- Regarding the lack of statistical analyses: We explicitly mention throughout the manuscript how many independent experiments and data points went into the graphs (and their error bars). When differences we build our story on are obvious and have correspondingly small error bars, the statistical analysis would only show the obvious, too, and would not provide any deeper insights. For comparison, we have looked at LSA papers highlighted on the LSA welcome page and they followed the exact same policy, i.e. provide statistical analyses when it makes sense but not for every possible comparison that displays obvious differences.

- ChIP-seq analysis not conclusive: you must be referring to the first comment of reviewer #2, but this is about ChIP experiments and NOT ChIP-seq data.

- Proliferation assays: these are beyond the point (see response to comment #2 of reviewer #1).

- More analyses of clinical datasets: more seems always better, but it may not always provide a better understanding. The database we used happens to be the best or at least one of the best (see response to comment #5 of reviewer #1).

Reviewer #1:

We appreciate the overall positive comment "Although this finding is interesting" and would be happy to provide the complements regarding the statement "some of issues need to be investigated and explained".

Major Comments

1) *In this MS, the authors only used one cell line for their study. At least ERa positive two cell lines like MCF7 and T47D, and one ER negative cell line (as a negative control) should be involved in.* While we cannot dispute the fact that all experiments shown were done with a single cell line (MDA-MB-134), we have of course done many of the key experiments with other cell lines, too. We notably have data obtained with MCF7 cells showing that CREB1 overexpression, mutation or knock-down affects ERa recruitment to target sites. Moreover, we have found that CREB1 affects the activity of a transfected ERa reporter the same way in ER-negative 293T cells as in the ones we used for the paper. Except for the fact that 293T are ER-negative cells, we do not think that doing ER-focused experiments in an ER-negative cell line, as a negative control, would be useful as we have included appropriate negative controls in all our experiments.

2) *Fig EV3, As we know, ERa is essential for ERa positive cell proliferation and mitochondrial membrane potential(MMP) is a key parameter of cell viability/survival, but the authors found that ERa KD increased MMP. So, cell proliferation assays should be involved in these cell lines.* No, we did not find that the ERa KD increases the MMP. The absolute values (and the means indicated across the different panels of Fig. EV3 cannot be compared between different panels (i.e. A/B/C). These are different cultures of cells with different knock-downs (with control shRNA versus ERa shRNA versus siCREB1). What can be compared in each case most reliably is minus and plus induction of apoptosis (C versus STS). Proliferation assays are beyond the point here, apart from the fact that KDs of ERa and CREB1 are not possible for such long-term assays, both ERa and CREB1 being essential.

Minor comments

1) *Statistical analysis should be involved in these bar graphs,*

Wherever appropriate, we are happy to add statistical analyses. However, when differences we build our story on are obvious and have correspondingly small error bars, the statistical analysis would only show the obvious, too, and would not provide deeper insights.

2) *Fig1b, the authors should explain why Vector control and CREB1S-133D group have the similar effects on LUC reporter assays.*

Not clear what this comment refers to. In the presence of cAMP (FI), endogenous CREB1 also gets phosphorylated/activated and may work as well as S133D. Why wild-type CREB1 gives a slightly higher increase is difficult to explain, but after all phosphorylated CREB1 might be slightly better than its mimic S133D. In any case, what's most relevant for this control experiment is the inhibitory effects of overexpressing the other CREB1 mutants.

3) *Fig 2a, c, E2 treatment, CREB1S-133D group even lower than Control? Please explain it.*

We did. We explicitly point it out and offer a speculative explanation for this finding (on page 6).

4) *Fig4, some of the WB film are very fuzzy, Please repeat them.*

For this experiment, a whole series of antibodies were necessary. And yes, some of the available ones may not be as good as one might hope. Nevertheless, we are convinced that the main conclusions are nicely supported by the blots as shown.

5) *TCGA database about breast cancer patients should be analyzed.*

Although the GOBO database may not be as well known as others, it actually has the highest numbers of useful samples for breast cancer. The number of cases are shown in all panels of our figures. Both TCGA and the Kaplan-Meier Plotter (<http://kmplot.com/analysis/>) actually have smaller numbers and therefore, by the time one looks at the relevant subsets of the data, results are far less convincing (even if the p values hold up).

Reviewer #2:

Thank you for the statements that "Overall, I had a very positive impression of the manuscript" and "Overall, I found this to be an informative paper, wherein the conclusions are supported by the data".

Major Criticisms

In Figure 3, ChIP fold recruitment of ER α to NF κ B, TFF1 and GREB1 is extremely highly variable in the vector control condition, ranging from 20-fold in 3A, to 8-fold in 3B to up to 400-fold in 3C. Since these are the same cells with the same conditions, this extremely high disparity indicates that experimental different or methodological difference may be the contributing factor. With such high variation, the rest of data might be suspect.

Yes we are aware of this issue, and in fact, we even pointed it out (page 8). With these biological experiments, fold recruitment of ER α is variable in our hands, and hence, the values themselves cannot be directly compared between different experiments, even if the experimental set-up is the same. What can and must be compared is the differences between different treatments and conditions within the same experiment, i.e. that shown in a single panel.

In Figure 7A, there are no statistical significance tests. The data presented seems from be from a single experiment. Would be much stronger point with some sort of significance test.

These are data from a single representative experiment. We could easily add statistics, but as pointed out in a response to a comment of reviewer #1, when the differences are visually obvious (and the error bars very small), there are no revelations to be expected from this type of analysis.

In general, the experiments shown in this paper have no significance testing. In the first seven figures, the only p value that can be seen is in Figure 7B comparing survival curves.

Again, as mentioned above, wherever appropriate, we are happy to add statistical analyses.

However, when differences we build our story on are obvious and have correspondingly small error bars, the statistical analysis would only show the obvious, too, and would not provide deeper insights. Moreover, we explicitly mention throughout the manuscript how many independent experiments and data points went into the graphs.

I think the authors are overselling the point about the protection from apoptosis provided by CREB1. Rather, if they should choose to make this point, they should do at least one other independent experiment to assay the antiapoptotic effects of CREB1, measurement of effects on BCL2 or the Caspase pathways.

We respectfully disagree with the statement that we are overselling the protection provided by CREB1. Indeed, we don't even mention that in the Title or the Abstract. Investigating how CREB1 protects against apoptosis is totally beyond the scope of this study.

Thank you for your recent correspondence regarding our decision on your manuscript entitled "ER α activity depends on interaction and target site corecruitment with phosphorylated CREB1". Two editors have re-assessed your work in light of your comments, and we additionally sought arbitrating advice on your work and the point-by-point response provided upfront from an additional expert working in the field.

We appreciate that you clarify that the reviewer comments you have received can be easily addressed, and that you already have data at hand to do so and to generalize your findings. The arbitrating expert thinks that the proposed revision would result in a robust study. We would thus like to invite you to submit a revised version of your manuscript. We encourage you to include the data obtained in different cell lines. To address the reviewers' comment regarding protection from cell death, we encourage you to include replicates for figure 7a as supplementary figure files.

Thank you for this interesting contribution to Life Science Alliance. We are looking forward to receiving your revised manuscript.

1st Revision – authors' response

2 May 2018

General revisions:

- We have adapted the manuscript format to LSA guidelines. Hence, many of the supplementary figures had to be renumbered.
- Most revisions were prompted by comments of the Editor and the reviewers and are discussed below. The exception to that is a motif analysis, which we have added in the Discussion, second paragraph of chapter "Molecular mechanism of", along with Fig S13 and the relevant section in Materials and Methods.
- As suggested by the Editor, we have added data obtained with other cell lines and revised Fig 7A to incorporate data from three independent experiments.

Original Editor's comments:

Importantly, they note a lack of generalisation and lack of inclusion of statistical analyses. They further think that the ChIP-seq analysis is non-conclusive and that several assays additional assay such as cell proliferation assays, inclusion of TCGA database analysis, and further proof of protection from apoptosis via CREB1 would be needed.

- Regarding the lack of statistical analyses: We explicitly mention throughout the manuscript how many independent experiments and data points went into the graphs (and their error bars). When differences we build our story on are obvious and have correspondingly small error bars, the statistical analysis would only show the obvious, too, and would not provide any deeper insights. For comparison, we have looked at LSA papers highlighted on the LSA welcome page and they followed the exact same policy, i.e. provide statistical analyses when it makes sense but not for every possible comparison that displays obvious differences. *This is what we have now done for the revision by adding statistical analyses to Figures 1, 2, 3, S2, and S3 (they all show what was to be expected).*
- ChIP-seq analysis not conclusive: this must be referring to the first comment of reviewer #2, but this is about ChIP experiments and NOT ChIP-seq data.
- Proliferation assays: these are beyond the point (see response to comment #2 of reviewer #1).
- More analyses of clinical datasets: The database we used happens to be the best or at least one of the best (see response to comment #5 of reviewer #1). *Nevertheless, we have now interrogated TCGA and show a side-by-side comparison for comparable datasets in the new Fig S12.*

Reviewer #1:

We appreciate the overall positive comment "Although this finding is interesting" and are happy to provide the complements regarding the statement "some of issues need to be investigated and explained".

Major Comments

1) In this MS, the authors only used one cell line for their study. At least ER α positive two cell lines like MCF7 and T47D, and one ER negative cell line (as a negative control) should be involved in.

Good point. Since we had these data all along, we are happy to include them now. We have found that CREB1 affects the activity of a transfected ER α reporter the same way in ER-negative 293T cells expressing exogenous ER α (new Fig S2). The new Fig S4 presents data obtained with MCF7 cells showing that CREB1 overexpression, mutation or knock-down affects ER α recruitment to target sites the same way as in MDA-MB-134 cells.

2) Fig EV3, As we know, ER α is essential for ER α positive cell proliferation and mitochondrial membrane potential(MMP) is a key parameter of cell viability/survival, but the authors found that ER α KD increased MMP. So, cell proliferation assays should be involved in these cell lines.

Note that Fig EV3 has now become Fig S10. No, we did not find that the ER α KD increases the MMP, but thank you for pointing out an issue that can lead to confusions. The absolute values (and the means indicated across the different panels of Fig S10 cannot be compared between different panels (i.e. A/B/C). These are different cultures of cells with different knock-downs (with control shRNA versus ER α shRNA versus siCREB1). What can be compared in each case most reliably is minus and plus induction of apoptosis (C versus STS). Proliferation assays are beyond the point here, apart from the fact that KDs of ER α and CREB1 are not possible for such long-term assays, both ER α and CREB1 being essential. We have now added a statement clarifying this in the legend to Fig S10.

Minor comments

1) *Statistical analysis should be involved in these bar graphs,*

Wherever appropriate, we have now added statistical analyses. See also general comments above about statistical analyses.

2) *Fig1b, the authors should explain why Vector control and CREB1S-133D group have the similar effects on LUC reporter assays.*

Unfortunately, it is not quite clear to us what this comment refers to. In the presence of cAMP (FI), endogenous CREB1 also gets phosphorylated/activated and may work as well as S133D. Why wild-type CREB1 gives a slightly higher increase is difficult to explain, but after all phosphorylated CREB1 might be slightly better than its mimic S133D. In any case, what's most relevant for this control experiment is the inhibitory effects of overexpressing the other CREB1 mutants.

3) *Fig 2a, c, E2 treatment, CREB1S-133D group even lower than Control? Please explain it.*

We did. We explicitly point it out and offer a speculative explanation for this finding (on pages 6/7).

4) *Fig4, some of the WB film are very fuzzy, Please repeat them.*

For this experiment, a whole series of antibodies were necessary. And yes, some of the available ones may not be as good as one might hope. Nevertheless, we are convinced that the main conclusions are nicely supported by the blots as shown.

5) *TCGA database about breast cancer patients should be analyzed.*

Thank you for prompting us to do that. We interrogated TCGA with the "Kaplan-Meier Plotter" and essentially came to the same conclusions (new Fig S12 and corresponding text in main manuscript). Please note that the GOBO database, even though it may not be as well known as others, actually has the highest numbers of useful samples for breast cancer. Hence, our original choice to focus on that one.

Reviewer #2:

Thank you for the statements that "Overall, I had a very positive impression of the manuscript" and "Overall, I found this to be an informative paper, wherein the conclusions are supported by the data".

Major Criticisms

In Figure 3, ChIP fold recruitment of ER α to NF κ B, TFF1 and GREB1 is extremely highly variable in the vector control condition, ranging from 20-fold in 3A, to 8-fold in 3B to up to 400-fold in 3C. Since these are the same cells with the same conditions, this extremely high disparity indicates that experimental different or methodological difference may be the contributing factor. With such high variation, the rest of data might be suspect.

Yes we are aware of this issue, and in fact, we even pointed it out (page 8). With these biological experiments, fold recruitment of ER α is variable in our hands, and hence, the values themselves cannot be directly compared between different experiments, even if the experimental set-up is the same. What can and must be compared is the differences between different treatments and conditions within the same experiment, i.e. that shown in a single panel. We have now added a statement clarifying this in the legend to Fig 3.

In Figure 7A, there are no statistical significance tests. The data presented seems from be from a single experiment. Would be much stronger point with some sort of significance test.

Thank you for pointing out this shortcoming. We have now incorporated the data from three independent experiments. The result is exactly the same and the differences are so obvious (with very small error bars) that statistical tests cannot provide further support.

In general, the experiments shown in this paper have no significance testing. In the first seven figures, the only p value that can be seen is in Figure 7B comparing survival curves.

Overall, this has been fixed, but please see also our response to a general comment of the Editor.

I think the authors are overselling the point about the protection from apoptosis provided by CREB1. Rather, if they should choose to make this point, they should do at least one other independent experiment to assay the antiapoptotic effects of CREB1, measurement of effects on BCL2 or the Caspase pathways.

We respectfully disagree with the statement that we are overselling the protection provided by CREB1. Indeed, we don't even mention that in the Title or the Abstract. Investigating how CREB1 protects against apoptosis is totally beyond the scope of this study.

3rd Editorial Decision

18 May 2018

Thank you for submitting your revised manuscript entitled "ER α activity depends on interaction and target site corecruitment with phosphorylated CREB1".

One of the original reviewers as well as the arbitrating advisory expert previously involved have re-assessed your work, and both support publication in Life Science Alliance.

REFEREE REPORTS

Reviewer #2 (Comments to the Authors (Required)):

I like the changed that were made. I believe that this paper will add to the understanding of the interactions of estrogen receptor with other signaling pathways.

Arbitrating expert (Comments to the Authors (Required)):

As an independent reviewer, I believe that the authors have adequately addressed the reviewers comments and the revised version of the paper is an improved manuscript. The findings are interesting and will be of relevance to the community.